# Insight into the Inter-Organ Crosstalk and Prognostic Role of Liver-Derived MicroRNAs in Metabolic Disease Progression

**DOI:** 10.3390/biomedicines11061597

**Published:** 2023-05-31

**Authors:** Bruno de Souza Goncalves, Avery Meadows, Duane G. Pereira, Raghav Puri, Sneha S. Pillai

**Affiliations:** Department of Surgery and Biomedical Sciences, Joan C. Edwards School of Medicine, Marshall University, Huntington, WV 25701, USA; desouzagonca@marshall.edu (B.d.S.G.); meadows336@marshall.edu (A.M.); pereirad@marshall.edu (D.G.P.); rpuri1@stu.k12.wv.us (R.P.)

**Keywords:** microRNA, biomarkers, non-alcoholic fatty liver disease, hepatic fibrosis, insulin resistance, cardiovascular diseases

## Abstract

Dysfunctional hepatic metabolism has been linked to numerous diseases, including non-alcoholic fatty liver disease, the most common chronic liver disorder worldwide, which can progress to hepatic fibrosis, and is closely associated with insulin resistance and cardiovascular diseases. In addition, the liver secretes a wide array of metabolites, biomolecules, and microRNAs (miRNAs) and many of these secreted factors exert significant effects on metabolic processes both in the liver and in peripheral tissues. In this review, we summarize the involvement of liver-derived miRNAs in biological processes with an emphasis on delineating the communication between the liver and other tissues associated with metabolic disease progression. Furthermore, the review identifies the primary molecular targets by which miRNAs act. These consolidated findings from numerous studies provide insight into the underlying mechanism of various metabolic disease progression and suggest the possibility of using circulatory miRNAs as prognostic predictors and therapeutic targets for improving clinical intervention strategies.

## 1. Introduction

Liver disease is a leading cause of mortality worldwide and constitutes a wide range of diseases with varied or unknown etiologies [1]. Globally, liver disease causes approximately 2 million deaths per year and, besides an increased risk of mortality, chronic liver disease poses a critical impact on the economy and quality of life indices [2]. While progress has been made in understanding the causes of liver disease and in developing treatments, there are still significant challenges to be elucidated. The prevalence of obesity has increased dramatically during recent decades and numerous studies have explored the association between obesity and liver dysfunction [3,4,5,6,7]. Obesity is associated with a spectrum of liver abnormalities, especially non-alcoholic fatty liver disease (NAFLD), which can progress to severe liver dysfunction and can cause metabolic syndrome (MetS) and comorbidities, including type 2 diabetes mellitus (T2DM), hypertension, hyperlipidemia, and cardiovascular disease (CVD), leading to increased mortality [8,9]. Hence, an understanding of the mechanism and mediators of various cellular functions and disease processes in the liver can provide a strong basis for future research and therapeutic applications.

MicroRNAs (miRNAs) constitute a class of short, non-coding RNAs with 19 to 25 nucleotides that are transcribed by RNA polymerase II; additionally, the sequences of most miRNAs are conserved among different species [10]. miRNAs serve as post-transcriptional regulators of gene expression and often have multiple targets within the same biological pathway; in fact, each target can be regulated by different miRNAs. Thus, the deregulation of various miRNA profiles safeguards the stability of biological systems by regulating essential biological processes such as metabolism, cell growth, differentiation, stress response, and tissue remodeling [11]. Evidence suggests that the understanding of the miRNA regulation of human genes and their involvement in various disease progression has been studied in detail over the past decade [12]. Of note, miRNAs exhibit remarkable stability in various body fluids and specific tissues and the expression of many miRNAs is specific to various physiological and pathological conditions [13]. miRNAs act as significant therapeutic targets in personalized medicine and provide an integrated readout of various biological conditions [14]. For example, microarray studies have facilitated the identification of numerous miRNAs that can serve as biomarkers [12].

Mounting evidence suggests that miRNAs regulate both physiological and pathological liver functions in almost all acute and chronic liver diseases [15,16,17]. The altered expression of miRNAs is associated with metabolic dysregulation, liver injury, liver fibrosis, and tumor development, making them attractive candidates for the diagnosis and treatment of liver diseases [16]. In recent years, increasing evidence has indicated the involvement of miRNAs in regulating cholesterol and lipid metabolism in the liver and their potential contribution to the development of metabolic disturbances and CVD [11]. As the localization of the tissue of origin of miRNAs is very difficult, and numerous miRNAs are secreted from multiple organs, it is a captivating challenge to explain the role of microRNA regulation associated with disease progression. Hence, the purpose of this review is to elucidate the mechanistic role of liver-derived miRNAs in hepatic lipid metabolism and the development of various pathologies. As shown in Figure 1, liver-derived miRNAs have been demonstrated to have multiple tissue/organ targets and modulate various metabolic pathways associated with diseases, such as NAFLD [18,19], insulin resistance [20], fibrosis [21], and CVD [11]. Furthermore, this review assesses the feasibility of detecting liver miRNAs in circulation, which is a prerequisite for their use as soluble biomarkers. Additionally, the review summarizes the evidence from epidemiological studies regarding the association of liver miRNAs in circulation and discusses the necessary steps for exploiting these molecules as possible therapeutic targets and prognostic predictors in future studies.

## 2. Biogenesis and Release of miRNA from the Liver

miRNA biogenesis entails several steps, including transcription, cleavage, and maturation [22]. They are first transcribed as long, variable-length hairpin precursor molecules, followed by processing via RNAse III protein Drosha and endonucleases to generate mature miRNAs [10]. The mature miRNA can then associate with an RNA-induced silencing complex, thereby engaging in post-transcriptional regulation [10]. This interaction between miRNAs and the complex results in the binding of miRNAs to specific mRNA sequences, interfering with translocation or reducing the stability of target mRNAs [23]. miRNAs possess partial complementarity to regulatory regions situated in the 3’ or, less commonly, in the 5’ untranslated region of target messenger RNAs [24]. The binding of miRNAs to their target mRNA leads to the suppression of mRNA translation or the promotion of mRNA degradation, which constitutes a rapid mechanism for fine-tuning gene expression [25].

Hepatic miRNA profiles exert a significant influence on the etiology of metabolic disorders through their regulatory impact on different targets in various organs [15,26]. The liver, being a multifaceted organ comprising various cell types, such as hepatocytes, hepatic stellate cells (HSCs), and immune cells, is subject to diverse factors that modulate the release of extracellular vesicles (EVs) containing miRNA by these cells. Factors such as alcohol, Toll-like receptor 9 (TLR9) ligands, or free fatty acids can influence the release of EVs, leading to their active involvement in intercellular communication not only within the liver but also with distant organs [27]. Exosomes originate from the endosomal system and are secreted by integrating multivesicular endosomes into the cell membrane [28]. Although the release of EVs may appear to be a simple process involving the budding off of vesicles from the plasma membrane, their trafficking and release entail multiple stages that can be regulated at various levels [29]. Upon reaching the target cells, EVs initiate internalization through dynamic interactions, involving a specific binding of membrane proteins to cell receptors [30]. Subsequently, membrane binding promptly progresses to fusion, wherein endocytosis serves as the main pathway for exosome uptake, thereby releasing their contents into the target cells and inducing alterations in cellular function by specific targets of miRNAs [30,31].

## 3. The Role of Liver-Derived miRNAs in Metabolic Regulation and Associated Pathologies

At first, it was hypothesized that miRNAs would only regulate gene expression in cells where they were endogenously produced. However, subsequent research has revealed that miRNAs can be secreted into extracellular fluids and transported to other cells, where they are capable of exerting their regulatory effects [32]. The recognition of miRNAs as signaling and coordinating entities between cells and tissues has established their potential utility as biomarkers of disease [33,34]. A broad array of miRNAs secreted from the liver have been established in terms of their function and cellular distribution in the liver as well as their potential role in inter-organ communication [15]. Hence, some major liver-derived miRNAs will be described in this review according to their metabolic regulatory functions as well as their possible potential in disease diagnosis and therapy.

### 3.1. Role of Liver-Derived miRNAs in NAFLD

NAFLD is one of the most common liver diseases in the world, especially in Western countries [35,36]. Studies have shown that NAFLD prevalence increased from 2.8% to 46%, and has been associated with the lifestyle of the population in urban centers, especially with obesity and metabolic disorders [35]. NAFLD is associated with various lipid and glucose abnormalities and may lead to the progression from steatosis to non-alcoholic steatohepatitis (NASH), fibrosis, and cirrhosis [11]. Presently, an invasive procedure remains necessary to confirm the diagnosis [19]. Several studies have strived for the identification of a serum/plasma miRNA signature of NAFLD; therefore, liver miRNAs have been implicated as promising non-invasive biomarkers and therapeutics in the NAFLD context (Table 1) [18,19,37]. In this section, we discuss the role of various liver-derived miRNAs in the pathogenesis of NAFLD and their dysregulation during various stages of disease progression.

miR-122 is one of the main liver-derived miRNAs, and is the most studied as well [38,39], making up to 70% of the organ’s miRNA content [40]. Studies have described that miR-122 has different targets regarding lipid and cholesterol metabolism, including fatty acid synthase (FAS), acetyl-CoA-carboxylase (ACC), stearoyl-CoA desaturase (SCD-1), sterol regulatory element binding proteins (SREBPs), and hepatocyte nuclear factor 6 (HNF6) [41,42]. It also regulates various liver-specific proteins involved in hepatocyte differentiation [42]. It is described that circulating levels of miR-122 are greatly increased in patients with NAFLD. This upregulation of miR-122 is positively correlated with the degree of inflammation and fibrosis and serves as a valuable indicator of NAFLD severity in various stages of liver dysfunction alongside traditional liver markers such as aspartate aminotransferase (AST) and cytokeratin-18 (CK18) [37,39,43].

The upregulated levels of miR-34a play an important role in the development of NAFLD/NASH [44]. It is not typically found in the serum of healthy patients but has been observed in patients with NAFLD [43,44]. Cermelli et al. conducted a study showcasing the elevation of miR-34a levels in the serum of patients as NAFLD/NASH progresses. These findings exhibit associations with liver enzyme levels, fibrosis stage, and inflammation activity [43]. This is a highly lipid-responsive miRNA present in the liver that modulates oxidative stress and lipid metabolism [45]. Indeed, the increased expression of miR-34a induces hepatocyte apoptosis by the acetylation of p53, showing a correlation with body mass index (BMI) in obese patients [46]. It has also been shown that miR-34 regulates sirtuin 1 (SIRT1), a NAD-dependent deacetylase, and prevents the activation of peroxisome proliferator-activated receptor alpha (PPARα) and the liver X receptor, proteins involved in fatty acid oxidation and energy metabolism [45,46,47]. In addition, human and murine studies have shown that miR-34a inhibited the liver secretion of very low-density lipoprotein (VLDL), causing steatosis when combined with hepatocyte nuclear factor 4 alpha (HNF4α) [48].

miR-21 is another one of the best-studied miRNAs in the liver and serum of patients with NAFLD/NASH [37,49]. Studies have shown that hepatic and serum miR-21 expression is increased in animal models [49,50] and patients [39] with NAFLD/NASH. Studies have shown that miR21 is involved with lipid accumulation in the liver through the regulation of various proteins including SREBP1 [51], 3-hydroxy-3-methylglutaryl-co-enzyme A reductase (HMGCR) [52], and fatty acid binding protein 7 (FABP7) [53]. Another mechanism of lipid accumulation by miR-21 is the inactivation of the WNT/β-catenin signaling pathway through its regulation of low-density lipoprotein (LDL) receptor-related protein 6 (LPR6) [54]. Calo et al. found that a liver-specific miR-21 removal decreased hepatic steatosis in mice fed a high-fat (HFD) diet through the regulation of transcription factors, including forkhead box protein O1 (FOXO1), forkhead box protein A2 (FOXA2), HNF4α, the signal transducer and activator of transcription 3 (STAT3), and insulin-induced gene 2 (INSIG2) [55]. Moreover, miR-21 can also target peroxisome proliferator-activated receptor alpha (PPAR-α) to increase the lipid and triglyceride levels in the liver, eventually leading to tissue damage [49]. Therefore, miR-21 has been shown to be an important miRNA in the context of liver disease progression, being a good marker of differentiating the late stages of liver dysfunction [37,39,43].

miR-15b is upregulated in both the liver and serum of patients with NAFLD and NAFLD experimental models [56]. It has been shown that increased levels of miR-15b in serum were associated with obesity and T2DM, suggesting its usefulness as a biomarker [57]. Studies have found that miR-15b is responsible for regulating proliferation and fibrinogenesis in hepatic stellate cells (HSCs) cells by regulating lysyl oxidase-like 1 (LOXL1) [58]. Furthermore, when miR-15 was overexpressed in hepatic cells, it caused an increase in triglyceride levels and a decrease in glucose consumption [56]. Therefore, these results suggest that the increase in miR15b levels may be related to an alteration in glucose and lipid metabolism, possibly increasing the risk of NAFLD.

Elevated levels of miR-33 have been observed in the serum of mice models with NAFLD, and a positive correlation between miR-33 levels in the liver of NAFLD patients and the advancement of the disease has also been reported [59,60]. It has been found to be an important regulator of lipid metabolism and insulin sensitivity by targeting the SREBPs and ATP binding cassette subfamily A member 1 (ABCA1) [59,61,62,63]. Studies using HSCs have shown that miR-33a was found to be correlated with the increased expression of type I collagen (Col1A1) and α-smooth muscle actin (αSMA) by transforming growth factor beta (TGFβ), suggesting its involvement in the progression of NAFLD to liver fibrosis [64]. Moreover, it has been demonstrated that the inhibition of miR-33 attenuates atherosclerosis progression, and the inhibition of these miRNAs was associated with the regeneration of damaged liver tissue [65,66].

miR-375 is significantly associated with NAFLD severity and its serum levels are highly upregulated in NAFLD [38,67]. It is important for the inflammatory process and glucose homeostasis and critical for helping β-cells to meet the increasing demand for insulin during insulin resistance [18,67]. miR-375 also targets the adiponectin receptor 2 (AdipoR2) protein, an important regulator of glucose and lipid metabolism, by activating the PPAR-α pathway, allowing it to reduce insulin sensitivity and steatosis [67].

Other miRNAs have also shown their regulatory role in NAFLD progression; miR-451 levels in serum are highly upregulated in the mild and severe stages of NASH [39]. It is a regulator of thyroid hormone responsive spot 14 (THRSP) expression, essential for de novo lipogenesis [68]. In addition, miR-190b is upregulated in NAFLD and regulates lipid metabolism and insulin sensitivity by targeting insulin-like growth factor 1 (IGF-1) and thrombospondin motifs 9 (ADAMTS9) [69]. Moreover, miR-181b is upregulated in the serum and liver and is involved in the progression of NAFLD by targeting SIRT1 and regulating lipid metabolism [70,71].

**Table 1 biomedicines-11-01597-t001:** Summary of the liver-derived miRNAs associated with non-alcoholic fatty liver disease.

miRNAs	Site of Action	Status in Disease	Targets	Physiological Effect	References
miR-122	Serum	Increased	FAS ACC SCD-1 SREBPs HNF6	Lipid metabolism and hepatocyte differentiation	[38,39,41,42]
miR-34a	Serum	Increased	P53 SIRT1 NAD-dependent deacetylase PPAR-α Liver x receptor HNF4α	Oxidative stress, lipid metabolism, hepatocyte apoptosis, and fatty acid oxidation	[43,44,45,46,47,48]
miR-21	Serum Liver	Increased Increased	SREBP1 HMGCR FABP7 LPR6 FOXA2 FOXO1 HNF4α STAT3 INSIG2 PPAR-α	Lipid metabolism	[37,39,49,50,51,52,53,54,55]
miR-15b	Serum L02 cells	Increased Increased	LOXL1	HSC activation, glucose metabolism, and lipid metabolism	[56,58]
miR-33	Serum Liver	Increased Increased	SREBPs ABCA1 TGFβ	Lipid metabolism and insulin sensitivity	[59,61,62,63,64]
miR-375	Serum	Increased	AdipoR2	Glucose metabolism, insulin sensitivity, and inflammation	[38,67]
miR-451	Serum	Increased	THRSP	Lipid metabolism	[39,68]
miR-190b	Liver	Increased	IGF-1 ADAMTS9	Lipid metabolism	[69]
miR-181b	Serum Liver	Increased Increased	SIRT1	Lipid metabolism	[70]

Abbreviations: Non-alcoholic fatty liver disease (NAFLD), fatty acid synthase (FAS), acetyl-CoA-carboxylase (ACC), stearoyl-CoA desaturase (SCD-1), sterol regulatory element binding proteins (SREBPs), hepatocyte nuclear factor 6 (HNF6), sirtuin 1 (SIRT1), hepatocyte nuclear factor 4α (HNF4α), 3-hydroxy-3-methylglutaryl-co-enzyme A reductase (HMGCR), fatty acid binding protein 7 (FABP7), LDL receptor-related protein 6 (LPR6), forkhead box protein A2 (FOXA2), forkhead box protein O1 (FOXO1), signal transducer and activator of transcription 3 (STAT3), insulin-induced gene 2 (INSIG2), lysyl oxidase-like 1 (LOXL1), ATP binding cassette subfamily A member 1 (ABCA1), transforming growth factor beta (TGF-β), adiponectin receptor 2 (AdipoR2), peroxisome proliferator-activated receptor alpha (PPAR-α), thyroid hormone responsive spot 14 (THRSP), insulin-like growth factor 1 (IGF-1), and thrombospondin motifs 9 (ADAMTS9).

### 3.2. Role of Liver-Derived miRNAs in Insulin Resistance

A sedentary lifestyle and access to high-energy foods can contribute to insulin resistance and T2DM [72]. Diabetes is an incredibly common metabolic disorder, affecting as much as 8% of the U.S. population, and it affects organs all over the body [73]. There are two types of diabetes: in type 1 diabetes mellitus (T1DM), the immune system attacks and kills the pancreatic β-cells that produce insulin, while in T2DM, insulin resistance develops [74]. Of the two types, type 2 makes up 95% of cases [75]. Studies demonstrate that the relationship between NAFLD, insulin resistance, and obesity is often intertwined with the association of different organs [35,76,77], so liver-derived miRNAs can also be used as biomarkers of insulin resistance (Table 2).

While it is primarily seen as an indicator of NAFLD, miR-122 can also be used as a biomarker for insulin resistance [20]. A study using human plasma samples demonstrated that high levels of miR-122 are related to MetS and T2DM [78]. In the same study, the inhibition of miR-122 in mice using an antagomiR led to several genetic downregulations of proteins involved in lipid metabolism, such as ATP citrate lyase (ACLY), microsomal triglyceride transfer protein (MTTP), and SREBP1 [78]. In addition, another study confirmed that obese patients had more than three times the level of circulating miR-122 compared to healthy controls [20]. Another study found that an increased level of miR-122 leads to a decrease in the expression of the protein IGF-1 receptor, suggesting that it is a target of miR-122 and is able to attenuate T2DM by increasing insulin sensitivity in the liver [79]. High levels of miR-122 have been associated with insulin resistance and diabetes in different tissues and cell lines, such as subcutaneous adipose tissue and pancreatic β-cells [80,81]. An overexpression of miR-122 in white adipose tissue has been observed in mice with a high-fat diet, and antagomir-122 treatment attenuated the miR-122 levels in white adipose tissue, liver, and plasma [82]. High serum levels of miR-122 were also demonstrated to be related to child obesity, with or without T2DM being positively correlated with interleukin-6 (IL-6) and tumor necrosis factor-alpha (TNF-α) levels [83].

miR-34a is another liver-derived miRNA that plays a role in insulin resistance [76]. In murine diet-induced obesity models, miR-34a expression was greatly increased in the liver, and the concomitant increase in plasma circulation was associated with the severity of both NAFLD and diabetes mellitus [80]. In an HNF4α-dependent manner, miR-34a can regulate the gene expression of proteins involved in glucose and lipid metabolism [48]. Furthermore, the overexpression of miR-34a led to a decrease in HNF4α levels that in turn caused an increase in hepatic triglycerides [48]. In addition, it has been demonstrated that high levels of hepatic miR-34a by targeting ubiquitin-specific peptidase 10 (USP10) reduced its expression, promoting the development of NAFLD [69]. In addition, the decrease in long noncoding RNA myocardial infraction-associated transcript 2 (IncRNA Mirt2) expression resulted in insulin resistance and hepatic steatosis, proving that Mirt2/miR-34a-5p/USP10 is related to NAFLD [84]. miR-34a was also demonstrated to be overexpressed in the liver and pancreatic islet of aging mice with insulin resistance, evidencing a possible new biomarker of insulin resistance [85].

miR-802 plays a part in regulating hepatic insulin resistance [86]. miR-802 negatively regulates the protein hepatocyte nuclear factor 1 beta (HNF1β), the reduction in which increases the expression of the suppressor of cytokine signaling 1 and 3 (SOCS1 and SOCS3) proteins that in turn cause insulin desensitization and promote glucose production in the liver [87]. A study demonstrated that the overexpression of miR-802 in mice liver by a phosphatase and tensin homolog (PTEN) and phosphatidylinositol-3 kinase (PI3K) can decrease protein kinase B (Akt) phosphorylation levels, contributing to the development of obesity by metabolism abnormalities [88]. In neonatal rat ventricular myocytes, miR-802 negatively regulated insulin sensitivity by targeting heat shock protein 60 (HSP60), leading to reactive oxygen species (ROS) accumulation and cardiac insulin resistance [89]. The overexpression of miR-802 was observed in pancreatic islets as well as in the liver and white and brown adipose tissue of obese mice compared to the normal chow diet mice [90]. In pancreatic islets, miR-802 can target neurogenic differentiation 1 (NeuroD1) and frizzled class receptor 5 (Fdz5), impairing insulin transcription and insulin secretion, respectively, and performing an important role in the development of T2D [90]. High levels of miR-802 were observed in the serum of patients with T2DM, enabling its consideration as a possible biomarker for T2DM [91]. In a murine HFD model, the increase in miR-802 was correlated with the amount of glucose and insulin in the serum and appeared to decrease the levels of oxidative stress-related proteins such as catalase (CAT), glutathione peroxidase (GSH-Px), and superoxide dismutase (SOD) [92].

miR-126 participates in glucose homeostasis [93]. In rat insulinoma (INS-1) β cells, the role of miR-126in insulin resistance comes from its effect on the protein insulin receptor substrate 1 and 2 (IRS-1/2), which are involved in hepatic glucose homeostasis and insulin signaling [93]. In addition, the downregulation of miR-126 in adipose tissue was observed in obesity conditions related to insulin resistance [77]. Additionally, miR-126 can target C-C Motif Chemokine Ligand 2 (CCL2) directly in inflammatory conditions and the overexpression of miR-126 can attenuate CCL2 production in human adipocytes cells, regulating the inflammatory process reducing insulin resistance and making it an interesting target for T2D treatment [94]. A decrease in the circulating plasma levels of miRNA-126 in patients was found to be a significant predictor of diabetes mellitus [95,96]. Intriguingly, plasma disruption could be observed before the clinical manifestation of diabetes [95]. It can also be used as a biomarker of insulin resistance specifically because the serum levels of miR-126 are decreased in insulin-resistant type 2 [97]. In addition, the downregulation of miR-126 in murine livers was associated with increased hepatic lipid accumulation [98].

miR-143 affects glucose metabolism [72]. In obese mouse models, the overexpression of miR-143 was observed in different tissues, such as the pancreas and liver, demonstrating the involvement of miR-143 in obesity-associated insulin resistance [99]. The overexpression of miR-143 targets an oxysterol-binding protein-related protein (ORP8), leading to its downregulation and impairing insulin-induced AKT activation in obesity-associated diabetes [99]. miR-143 also directly targets mitogen-activated protein kinase 11 (MAPK11), which impairs gluconeogenesis by two of the involved proteins, phosphoenolpyruvate carboxykinase 1 (PCK1) and glucose-6-phosphatase catalytic (G6PC), in liver tissue, causing the dysregulation of glucose metabolism [100]. Serum levels of miR-143 were significantly increased in MetS patients compared to the control and were found to be associated with the risk of developing insulin resistance, suggesting the possible downregulation of circulating miR-143 to prevent this clinical condition [101].

Evidence shows that miR-30b induces insulin resistance by stressing the endoplasmic reticulum (ER) and suggests its prognostic potential as a biomarker of NAFLD [102]. In mice fed a high-fat diet, the expression of miR-30b was significantly upregulated in the liver and increased serum levels were positively associated with both hepatic steatosis and insulin resistance in human patients [102]. miR-30b exerts its effect on insulin resistance by targeting sarco(endo)plasmic reticulum Ca^2+^-ATPase 2b (SERCA2b), a calcium ATPase protein that brings calcium from the cytosol into the ER lumen [102]. The inhibition of SERCA2b due to the overexpression of miR-30b contributes to the impairment of insulin sensitivity [102]. Additionally, the increase in miR-30b is also related to cytokine-mediated β-cell dysfunction, in MIN6 cells, by targeting NeuroD1 [103]. A recent study demonstrated the association between miR-30b and lipid metabolism in NAFLD [104]. In this study, they showed that miR-30b can inhibit the triglyceride content and fatty acid deposition in Huh-7 cells by targeting peroxisome proliferator-activated receptor gamma coactivator 1-alpha (PPARGC1A) genes, regulating the expression of glucose transporter protein type 1 (GLUT1), PPAR-α, and SREBP-1 [104]. Studies have also demonstrated that miR-30b was upregulated in the serum of patients with T2DM [105] and upregulated in the liver samples of obese patients with NAFLD [106].

Studies have shown that miR-23a affects insulin resistance, making it an interesting biomarker for T2DM [107,108]. In a rat model of diabetes, low levels of miR-23a in the liver targeted NIMA-related kinase 7 (NEK7), leading to its upregulation to activate pyroptosis caused by NLR Family Pyrin Domain Containing 3 (NLRP3) activation [107]. miR-23a/b is also able to upregulate SREBP and FAS, leading to an increase in hepatic lipid accumulation in the liver of mice fed a HFD [109]. In human pancreatic islets and β-cells, proinflammatory cytokines downregulate the expression of inhibitors of miRNAs, such as miR-23a/b, leading to an increase in the expression of proapoptotic proteins (DP5 and PUMA) and consequent β-cell apoptosis [110]. Low levels of miR-23a-3p were observed in rat diabetic models [107] and the serum of patients with T2DM [107,108]. In individuals with impaired glucose tolerance, the plasma levels of miR-23a-3p were lower in comparison to those with normal glucose tolerance [111]. Considering these findings, miR-23a becomes interesting for predicting insulin resistance and the development of diabetes.

Other liver-derived miRNAs have also been described as correlated with insulin resistance. miR-499-5p affects insulin signaling and it has been found to be decreased in the liver [75] and circulation during insulin resistance [112]. Reduced miR-499-5p in the murine liver contributes to insulin resistance by targeting PTEN and consequently inhibiting glycogen production and the PI3K/AKT/glycogen synthase kinase (GSK) signaling pathway, indicating involvement in the progression of T2DM [75]. In addition, the overexpression of miR-499-5p can protect against insulin resistance [75]. miR-103 and miR-107 levels are increased in NASH and NAFLD pathologies and these conditions are associated with the development of diabetes [94]. In this case, the high expression of miR-103 and miR-107, in liver and adipose tissue, can regulate hepatic insulin sensitivity by targeting caveolin-1 (CAV-1), a crucial regulator of insulin resistance [113]. High levels of miR-103 have been observed in the serum of patients with NAFLD and have been shown to be involved in the insulin resistance of these patients [114]. miR-107 is also upregulated in mice liver and T2DM patient serum during insulin resistance [115].

**Table 2 biomedicines-11-01597-t002:** Summary of the liver-derived miRNAs associated with insulin resistance.

miRNAs	Site of Action	Status in Disease	Targets	Physiological Effects	References
miR-122	Plasma Serum Liver SC Pancreatic β cells WAT	Increased Increased	ACLY MTTP SREBP-1 IGF-1R	Lipid metabolism, insulin sensitivity, and insulin resistance	[78,79,80,81,82,83]
miR-34a	Plasma Liver Pancreatic islet	Increased Increased	HNF4α USP10	Glucose metabolism, lipid metabolism, and insulin resistance	[48,80,84,85]
miR-802	Serum Liver Pancreatic islet WAT BAT	Increased Increased	HNF1β HSP60 NeuroD1 Fzd5	Glucose metabolism, insulin resistance, insulin metabolism, and oxidative stress	[87,89,90,91]
miR-126	Plasma Serum Liver INS-1 β cells	Decreased Decreased	IRS-1 IRS-2 CCL2	Insulin resistance, glucose metabolism, and lipid metabolism	[93,94,95,96,97]
miR-143	Serum Liver Pancreas	Increased Increased	ORP8 MAPK11	Glucose metabolism and insulin resistance	[99,100,101]
miR-30b	Serum Liver MIN6 cells	Increased Increased	SERCA2b PPARGC1A NeuroD1	Endoplasmic reticulum stress, insulin sensitivity, and lipid metabolism	[102,103,104,105,106]
miR-23a	Serum Liver Pancreatic islets β-Cells	Decreased Decreased	NEK7 SREBP-1 FAS DP5 PUMA	Pyroptosis, lipid metabolism, inflammatory process, and insulin resistance	[107,108,109,110]
miR-499-5p	Plasma/serum Liver	Decreased Decreased	PTEN PI3K/AKT/ GSK	Insulin resistance	[75,112]
miR-103	Serum Liver Adipose tissue	Increased Increased	Caveolin-1	Insulin sensitivity	[113,114]
miR-107	Serum Liver Adipose tissue	Increased Increased	Caveolin-1	Insulin sensitivity	[113,115]

Abbreviations: White adipose tissue (WAT), ATP citrate lyase (ACLY), microsomal triglyceride transfer protein (MTTP), sterol regulatory element-binding protein 1 (SREBP-1), insulin-like growth factor 1 receptor (IGF-1R), hepatocyte nuclear factor 4α (HNF4α), ubiquitin-specific peptidase 10 (USP10), brown adipose tissue (BAT), hepatocyte nuclear factor 1 beta (HNF1β), heat shock protein family D (HSP60), neurogenic differentiation 1 (NeuroD1), frizzled class receptor 5 (Fdz5), insulin receptor substrate 1 (IRS-1) and 2 (IRS-2), C-C motif chemokine ligand 2 (CCL2), oxysterol-binding protein-related protein 8 (ORP8), mitogen-activated protein kinase 11 (MAPK11), sarco(endo)plasmic reticulum Ca^2+^-ATPase 2b (SERCA2b), peroxisome proliferator-activated receptor gamma coactivator 1-alpha (PPARGC1A), NIMA-related kinase 7 (NEK7), sterol regulatory element-binding protein-1 (SREBP-1), fatty acid synthase (FAS), p53-upregulated modulator of apoptosis (PUMA), and phosphatase and tensin homolog (PTEN).

### 3.3. Role of Liver-Derived miRNAs in Hepatic Fibrosis

Hepatic fibrosis is a complex and inevitable process of many chronic liver diseases, such as NAFLD and NASH [116]. The mechanism of hepatic fibrosis is extremely complex, covering large numbers of molecular events [117]. Liver fibrosis is defined as a heightened concentration of extracellular matrix (ECM) material in the liver; the majority of the matrix material is produced by HSCs, which lose their lipids after substantial inflammation and transform into myofibroblast-like cells where they secrete excessive amounts of ECM into injured sites in the liver [118]. Evidence has shown that miRNAs are involved in the complex process of fibrosis development [21]. Thus, miRNA interference, as a new mechanism for regulating gene transcription levels, can be used as a biomarker for liver fibrosis and its progression (Table 3). This section will describe the dysregulation of various miRNAs during hepatic fibrosis and consolidates the information from studies showing their progressive changes during different stages of fibrosis development.

miR-200 has been shown to be overexpressed in the serum during liver fibrosis in clinical studies and murine models [119] and it has a significant role regarding the severity of liver fibrosis [120]. Murakami et al. showed that the upregulation of miR-200 is positively correlated with the degree of inflammation and fibrosis and serves as a valuable indicator of fibrosis severity in patients with hepatitis [121]. It has also been described that the miR-200 family has multiple targets, each of which can affect different aspects of liver fibrosis, such as SIRT1, matrix metalloprotein-2 (MMP-2), Col1A1 production, protein kinase B in the PI3K/AKT pathway by targeting the transcriptional regulator friend of Gata 2 (FOG2), and mitogen-activated protein kinases (MAPK), which can exacerbate liver fibrosis [119,120,122].

It has been shown that the miR-30 family is downregulated in different stages of liver fibrosis, however, no correlation has been found between the expression levels and fibrosis progression [123,124,125]. It has been described that miR-30 has an important role in fibrosis pathogenesis. The overexpression of miR-30 has been found to ameliorate Beclin1-mediated autophagy, thus attenuating liver fibrosis in in vitro and in vivo models [123]. Autophagy in liver fibrosis provides a steady source of free fatty acids (FFAs) that contribute to the activation of HSCs [126]. It is well known that activated HSCs play a pivotal role in the development of liver fibrosis, being responsible for the excessive accumulation of ECM proteins in the liver [127]. Another study found that it contributes to the amelioration of liver fibrosis by interfering with epithelial-mesenchymal transition (EMT) by regulating snail family translational repressor 1 (SNAI1), a zinc finger transcription factor protein involved in EMT [125,128].

miR-146a exhibits downregulation within the liver during the occurrence of liver fibrosis, whereas it is found to be overexpressed in the serum [129,130,131,132]. Furthermore, investigations have established a negative correlation between diminished levels of miR-146a in the liver and heightened levels of this miRNA in the serum of individuals across varying stages of liver fibrosis [129,132]. Consequently, miR-146a emerges as a promising indicator for assessing the severity of fibrosis progression. It is a negative regulator of immune and inflammatory responses and is involved in the NF-κB pathway through the two known targets, tumor necrosis factor receptor-associated factor 6 (TRAF6) and interleukin-1 receptor-associated kinase 1 (IRAK1) [129,130,133]. It has been described that the downregulation of miR-146a in liver fibrosis promotes the activation of HSCs and the increase in collagen through the regulation of TGF-β1 [131,134]. It has also been shown that the overexpression of miR-146a can suppress the activation and proliferation of HSCs by targeting proto-oncogene Wnt-1 (WNT1) and proto-oncogene Wnt-5a (WNT5a) and, consequently, the effectors α-smooth muscle actin (α-SMA) and Col1A1 [131].

miR-140-3p is another miRNA that affects the proliferation of HSCs [135]. Its levels are elevated in both cellular and serum contexts, rendering it a valuable biomarker for detecting both early and late-stage fibrosis [136,137]. It has been described that miR-140-3p plays a role in the proliferation and fibrogenesis of HSCs by regulating PTEN protein, a tumor suppressor protein in the PTEN/AKT pathway that regulates liver fibrosis, indicating that miR-140-3p may be a potential novel molecular target for liver fibrosis [135].

It has been shown that other miRNAs also play important roles in liver fibrosis. miR-34 members are upregulated in liver fibrosis and could be related to lipid metabolism by targeting acyl-CoA synthetase long-chain family member 1 (ASCL1) and fibrosis progression via the TGF-β and SIRT1/p53 pathway [76,138,139,140,141]. Moreover, members of the miR-29 family demonstrate a downregulation pattern in animal models of hepatic fibrosis and exhibit stage-dependent downregulation in plasma samples obtained from patients with fibrosis. Consequently, this miRNA emerges as a robust indicator of the severity of the progression of fibrosis [142]. miR-29 acts through signaling pathways involving TGF-β, NF-κB, and PI3K/AKT [142,143]. miR-101 is also downregulated in the fibrotic liver as well as in the activated HSCs and injured hepatocytes in the process of liver fibrosis, acting as suppressors of TGFβ signaling by targeting TGF-β receptor I (TβRI) and its transcriptional activator Kruppel-like factor 6 (KLF6) during liver fibrogenesis [144]. It has also been shown that the downregulation of miR-101 is negatively correlated with the degree of fibrosis and cirrhosis and serves as a valuable indicator of fibrosis severity [145].

**Table 3 biomedicines-11-01597-t003:** Summary of the liver-derived miRNAs associated with hepatic fibrosis.

miRNAs	Site of Action	Status in Disease	Targets	Physiological Effect	References
miR-200	Serum L02 cells	Increased Increased	SIRT1 MMP-2 PI3K/AKT FOG2 MAPK	Extracellular matrix protein accumulation	[119,120,121]
miR-30	Liver HSC cells	Decreased Decreased	Beclin1 SNAI1	HSC activation	[123,124,125,128]
miR-146a	Liver HSC cells Serum	Decreased Decreased Increased	TRAF6 IRAK1 TGF-β1 WNT1 WNT5a	Immunological response, inflammation, and HSC activation	[129,130,131,132]
miR-140-3p	Liver HSC cells Serum	Increased	PTEN	HSC activation	[136,137]
miR-34a	Liver HSC cells	Increased Increased	ASCL1 TGF-β SIRT1/p53	Lipid metabolism	[138,139,140,141]
miR-29	Serum Liver HSC cells	Decreased Decreased Decreased	TGF-β NF-κB PIK3/AKT	Extracellular matrix protein accumulation and HSC activation	[142,143]
miR-101	Liver HSC cells	Decreased Decreased	TβRI KLF6	HSC activation	[144,145]

Abbreviations: sirtuin 1 (SIRT1), matrix metalloprotein-2 (MMP2), phosphatidyl-inositol 3-kinase/protein kinase B (PI3K/Akt), friend of Gata 2 (FOG2), hepatic stellate cells (HSCs), snail family translational repressor 1 (SNAI1), tumor necrosis factor receptor associated factor 6 (TRAF6), interleukin-1 receptor-associated kinase 1 (IRAK1), transforming growth factor beta 1 (TGF-β1), proto-oncogene Wnt-1 (WNT1), proto-oncogene Wnt-5a (WNT5a), phosphatase and tensin homolog (PTEN), acyl-CoA synthetase long-chain family member 1 (ASCL1), nuclear factor kappa-light-chain-enhancer of activated B cells (NF-κB), TGF-β receptor I (TβRI), and Kruppel-like factor 6 (KLF6).

### 3.4. Role of Liver-Derived miRNAs in Cardiovascular Diseases

CVDs, including myocardial infarction, atherosclerosis, and stroke, remain the leading cause of death worldwide, an estimated 32% of all deaths, resulting in approximately 17.9 million deaths per year [146,147]. Despite mainly affecting the liver, NAFLD is considered a disease with systemic effects, being a risk factor for the development of CVD which is considered the main cause of death of patients with NAFLD, representing one-third of the total mortality [148,149]. Insulin resistance, altered lipid metabolism, immunological imbalance, and oxidative stress may increase the risk of developing CVD in NAFLD [148,149]. In this context, recent findings demonstrated that liver-derived miRNAs, detected in the circulation, liver, heart, macrophages, and aorta, play an important role in the maintenance of glucose levels and lipid metabolism, and are thus considered possible markers for cardiovascular diseases (Table 4) [11].

miR-122 is described as one of the major liver-derived miRNAs capable of affecting the cardiovascular system [11]. In addition to being the miRNA predominantly expressed in the liver, it is closely correlated with cholesterol and fatty acid metabolism and may have a direct effect on the risk of CVD [11]. Through knockdown studies, miR-122 was shown to indirectly regulate the expression of genes involved in lipid and cholesterol synthesis in the liver, such as 3-hidroxi-3-metilglutaril-CoA synthase 1 (HMGCS1), 3-hidroxi-3-metilglutaril-coenzima A reductase (HMGCR), and MTTP [150,151,152,153]. These regulatory functions of miR-122 significantly reduce plasma cholesterol and triglyceride levels and also increase the oxidation of hepatic fatty acids, which leads to the development of cardiovascular disease [128]. Studies demonstrate a positive correlation between increased plasma levels of miR-122 and the development of heart diseases and the formation of atherosclerotic plaques [154,155]. In heart tissue, high levels of miR-122 directly target the Hand2 transcription factor and increase mitochondrial fission by dynamin-related protein-1 (Drp1), leading to cardiomyocyte apoptosis heart failures [156].

The most studied miRNA when it comes to cholesterol metabolism is miR-33a/b, located in the introns of the genes encoding the sterol regulatory element-binding proteins, SREBP2 and SREBP1, respectively [150,153]. miR-33a/b inhibits the expression of cholesterol transporters in macrophages present in atherosclerotic lesions, ATP binding cassette transporter A1 (ABCA1) and ABCG1, promoting the accumulation of cholesterol and lipids in the aorta, reducing their efflux from them, and consequently leading to the progression of atherosclerotic plaque formation [157,158]. Studies show that anti-miR-33 therapy increases plasma high-density lipoprotein (HDL) levels, promotes cholesterol efflux, inhibits inflammatory processes, and reduces the progression of atherosclerosis [159,160,161]. In the liver, miR33a/b also targets genes involved in fatty acid β-oxidation, which can be indirectly involved in the development of CVD, including sirtuin 6 (SIRT6), carnitine palmitoyltransferase 1A (CPT1A), and activated protein kinase (AMPK), as well as genes involved in insulin signaling, such as insulin receptor 2 (IRS2) [63]. In summary, the increased levels of plasma miR-33 can be associated with the development of CVD [162], and hence, it may act as a biomarker and target for the treatment of atherosclerosis and lipid metabolism-related diseases.

Another miRNA involved in cholesterol metabolism and transport in the liver and macrophages within atherosclerotic plaques is miR-144 [163,164]. It has been described that miR-144 is up-regulated in hepatocytes and hepatocyte macrophages during obesity and insulin resistance, culminating in lipid accumulation and a decreased antioxidant response [165,166]. The bile acid receptor, farnesoid nuclear receptor X (FXR) in the liver, and liver receptor X, present in macrophages, positively regulate the expression of miR-144, which targets the ABCA1 receptor (involved in cholesterol efflux), leading to its inhibition and consequent lipid accumulation and low HDL levels [163,164]. Being a post-transcriptional regulator of ABCA1 in the liver and in macrophages within atherosclerotic plaques, silencing of miR-144 was able to increase plasma HDL levels, enhance reverse cholesterol transport, and protect against atherosclerosis in male mice [167]. Furthermore, a study using agomir, to overexpress miR-144, demonstrated a decrease in ABCA1 receptor levels in the liver and aorta and a consequent decrease in HDL levels and reverse cholesterol transport, leading to the accelerated progression of atherosclerosis [168]. Thus, increased plasma levels of miR-144 are related to the severity of CVD [169].

miR-223 is expressed in different cell types and targets a wide variety of genes as it is able to regulate the biological processes involved in metabolic homeostasis, such as cholesterol biosynthesis and inflammatory processes [11,170,171]. Liver-derived miRNA-223 is described to inhibit cholesterol biosynthesis through the direct inhibition of two specific enzymes, methylsterol monooxygenase 1 (MSMO1) and sterol enzyme 3-hydroxy-3-methylglutaryl-CoA synthase 1 (HMGCoA), and the repression of scavenger receptor BI (SCARB1) [171]. It also reduces HDL-cholesterol uptake through repression of the scavenger receptor B1 and increases cholesterol efflux through positive regulation of the ABCA1 receptor, which may impact the development of cardiovascular disease [171]. miRNA-233 is also described to modulate different signaling pathways, such as the PI3K/PKB, NF-kb, and MAPK pathways affecting cell proliferation, migration, and apoptosis in different cell types, such as hepatocytes and cardiomyocytes [170]. However, miR-223 in the heart can directly target the apoptosis repressor with the caspase recruitment domain (ARC), leading to cardiac hypertrophy and heart failure [172]. Therefore, high plasma levels of miRNA-223 have been shown to be closely correlated with CVD pathology, including myocardial ischemia and atherosclerosis, acting as a potential biomarker and therapeutic target of these pathologies [173,174].

The miR-30 family consists of five members with ubiquitous expression, with miR-30c being the main one in relation to fibroblasts and cardiomyocytes [175]. miR-30c can regulate different processes such as apoptosis, cell proliferation, and the processes related to cardiac fibrosis and hypertrophy by disturbing the mitochondrial function in the heart [175]. In addition, in vitro studies have demonstrated some possible targets for miR-30c, such as plasminogen-activator inhibitor-1 (PAI-1)—important in the activation of matrix metalloproteases, p53, metastasis-associated gene-1 (MTA1), and MTTP, related to apoptosis, cell proliferation, and lipid metabolism, respectively [176,177,178,179]. MTTP is important in the assembly of lipoproteins and consequently in the maintenance of lipid metabolism, cholesterol levels, and the development of atherosclerosis, proven by aortic analysis [180]. Furthermore, the overexpression of miR-30c in cardiomyocytes resulted in dilated cardiomyopathy [11]. The severity of coronary lesions and elevated PAI-1 and vitronetin levels were found to be negatively associated with the reduced plasma levels of miR-30c, which may serve as a diagnostic biomarker for coronary heart disease [181].

miRNA-128 is expressed in different tissue types, such as the brain, heart, and liver, playing an important role in liver injury and cardiovascular disease [182]. High levels of miR-128 can be observed in hepatocyte-derived EVs from the plasma of patients with NASH compared to controls, and in the circulation of patients with NAFLD, which may consequently lead to cardiovascular dysfunction [183]. Additionally, the upregulation of miR-128 was demonstrated to mediate heart failure by the direct inhibition of Axin1, an inhibitor of the Wnt/β-catenin signaling pathway [184]. The downregulation of miR-128 was also demonstrated to attenuate Ang II-induced apoptosis, autophagy, and oxidative stress by targeting PIK3R1/Akt/mammalian target of rapamycin complex 1 (mTORC1) and/or the SIRT1/p53 pathways in a mouse heart model [182]. In addition, miR-128 is involved in hepatic lipid metabolism and decreasing cholesterol efflux by direct inhibition of the ABCA1 transporter, which may affect the cardiovascular system [185]. Therefore, miR-128 can be considered a potential biomarker and therapeutic target for heart failure and CVDs.

miR-148 may be related to the development of atherosclerosis in association with hepatic lipid metabolism [11]. Studies have demonstrated that miR-148 acts as a negative regulator of LDL receptor (LDLR) expression and the inhibition of this miRNA could lead to LDL clearance, decreasing plasma cholesterol levels and regulating LDL-C uptake by the SREBP1-mediated pathway [186,187]. The same happens with the ABCA1 transporter, where miR-148 inhibition increases transporter expression in the liver, controlling cholesterol efflux [186]. In addition, miR-148 also has other targets involved in energy and lipid metabolism, such as CPT1a, AMPK, and salt-inducible kinase 1 (SIK1) [187]. Therefore, especially through the effect on lipid metabolism, high plasma levels of miR-148 seem to contribute to the progression of atherosclerosis and its antagonism can attenuate the atherogenic process [186]. Serum levels of miR-148 were also demonstrated to be increased in patients with MetS, a risk factor for cardiovascular disease [188]. Through atherosclerotic lesion and atherosclerotic plaque analysis, miR-148 was demonstrated to regulate macrophage-related lipoprotein metabolisms, such as cholesterol efflux, and inflammatory processes affecting atherosclerosis progression [189].

Other miRNAs derived from the liver have also been described to have impacts on the risk of cardiovascular diseases. A high expression of miR-143 was demonstrated in the plasma of patients with NAFLD and coronary artery disease (CAD), demonstrating that NAFLD can be a risk factor for the development of CAD as well [190]. miR-143 can directly target the Ets LiKe gene 1 (Elk1), reducing its expression and inhibiting cardiac smooth muscle differentiation [191]. Liver-derived miR-24 is also implicated in the risk of cardiovascular disease, especially atherosclerosis [192]. In this context, the high expression of miR-24 was demonstrated in HFD mice livers, human hepatocytes incubated with fatty acids, and by directly repressing insulin-induced gene 1 (Insig1), an inhibitor of lipogeneses, which can lead to lipid accumulation and atherosclerosis [192]. Therefore, miR-24 inhibition could be an interesting therapeutic target in the treatment of NAFLD and cardiovascular diseases linked to liver damage [192].

**Table 4 biomedicines-11-01597-t004:** Summary of the liver-derived miRNAs associated with cardiovascular diseases.

miRNAs	Site of Action	Status in Disease	Targets	Physiological Effects	References
miR-122	Plasma Liver Cardiac rissue	Increased	HMGCS1 HMGCR MTTP HAND2	Lipid and cholesterol synthesis and apoptosis	[150,151,152,153,154,155,156]
miR-33	Plasma Hepatocyte Cell lines Liver Aorta	Increased	ABCA1 ABCG1 SIRT6 CPT1A AMPK IRS2	Cholesterol transport and lipid metabolism	[63,157,158,162]
miR-144	Plasma Macrophages Liver Aorta	Increased	ABCA1	Lipid metabolism and antioxidant response	[163,164,167,168,169]
miR-223	Plasma Liver Heart	Increased	HMGCoA MSMO1 SCARB1 ABCA1	Cholesterol metabolism, inflammatory process, and hypertrophy	[171,172,173,174]
miR-30c	Plasma Heart Cell lines	Increased Increased	PAI-1 p53 MTA1 MTTP	Fibrosis, hypertrophy, apoptosis, and lipid metabolism	[176,177,178,179,180,181]
miR-128	Plasma Heart Cardiac cells	Increased	Axin1 PI3K/Akt/mTORC1 SIRT1/p53 ABCA1	Oxidative stress and lipid metabolism	[182,183,184,185]
miR-148	Plasma Serum Liver Hepatic cells Aorta	Increased	LDLR SREBP1 ABCA1 CPT1A AMPKa1 SIK1	Lipid metabolism, lipid transport, inflammatory process, and cholesterol efflux	[186,187,188,189]
miR-143	Plasma Cardiac cells	Increased	Elk-1	Cell differentiation	[190,191]
miR-24	Liver Human Hepatocytes	Increased	Insig1	Lipid metabolism	[192]

Abbreviations: 3-hidroxi-3-metilglutaril-CoA sintase 1 (HMGCS1), 3-hidroxi-3-metilglutaril-coenzima A redutase (HMGCR), microsomal triglyceride transfer protein (MTTP), heart and neural crest derivatives expressed 2 (HAND2), ATP binding cassette subfamily A member 1 (ABCA1), ATP binding cassette subfamily G member 1 (ABCG1), sirtuin 6 (SIRT6), carnitine palmitoyltransferase 1A (CPT1A), AMP-activated protein kinase (AMPK), insulin receptor substrate 2 (IRS2), 3-hydroxy-3-methylglutaryl-CoA synthase 1 (HMGCoA), methylsterol monooxygenase 1 (MSMO1), scavenger receptor BI (SCARB1), plasminogen-activatorinhibitor-1 (PAI-1), metastasis-associated gene-1 (MTA1), phosphatidyl-inositol 3-kinase/protein kinase B/mammalian target of rapamycin complex 1 (PI3K/Akt/mTORC1), sirtuin-1/p53 (SIRT1/p53), LDL receptor- (LDLR), sterol regulatory element-binding protein 1 (SREBP1), 5′ adenosine monophosphate-activated protein kinase a1 (AMPKa1), salt-inducible kinase 1 (SIK1), Ets LiKe gene 1 (Elk1), and insulin-induced gene 1 (Insig1).

## 4. Conclusions

In pathophysiological processes, dysfunctional hepatic metabolism plays a critical role, and miRNAs have emerged as key players in these conditions. These miRNAs modulate the expression of several genes, thus promoting disease development through different cellular pathways. Early identification of diseases resulting from liver dysfunction is crucial, and biomarkers may play a vital role in their prognosis. The identification of potential markers can provide essential information for patient stratification and the development of individualized therapy. This review summarizes the involvement of liver-derived miRNAs in metabolic disease progression, delineating the communication between the liver and other tissues (Figure 2).

The extracellular vesicle-mediated organ crosstalk of miRNAs makes them promising candidates for therapeutic applications. miRNA-based novel drug delivery systems are one of the most significant therapeutic breakthroughs for next-generation drugs for the treatment of various ailments. The exploration of the origin and biological action of specific miRNAs will aid in future research and clinical trials. miRNAs act as an ideal biomarker and a reliable tool for clinical applications because of their easy accessibility, high specificity, and sensitivity. The databases and algorithms available in various bioinformatics tools can be utilized to predict miRNA regulatory targets and their corresponding implicated biological pathways. These miRNA data repositories will further assist future clinical research for the validation of novel therapeutics. Putative miRNA biomarkers will undoubtedly enable earlier diagnosis and expedited medical intervention strategies, which will expand the focus of human clinical trials. Hence, future clinical steps should be taken by figuring out the specific role and target of miRNAs in various pathological conditions to formulate the strategy of diagnosis, prognosis, and treatment.

Despite the existence of several challenges, recent advancements in technologies aimed at investigating miRNAs, including sequencing technologies and screening for miRNA mimic or inhibition, hold promise for the identification of novel targets that can be utilized in the development of diagnostics and therapeutics for liver disease. Thus, the clinical use of the miRNAs described in this review may significantly improve the development of new policies to improve the identification and management of patients having different diseases associated with liver dysfunction.

## Figures and Tables

**Figure 1 biomedicines-11-01597-f001:**
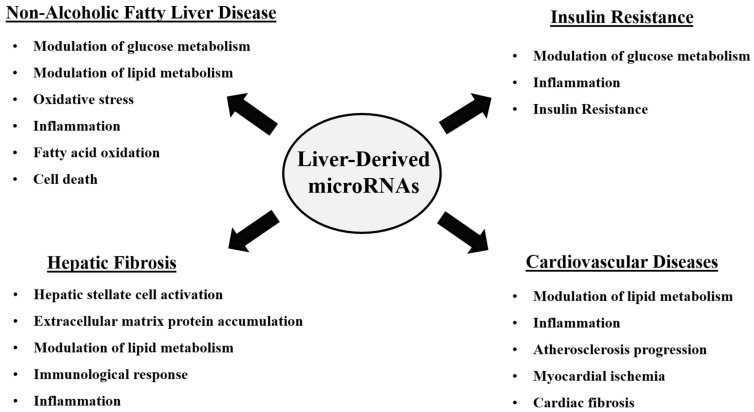
The role of liver-derived miRNAs in the regulation of various biological processes involved in metabolic disease development.

**Figure 2 biomedicines-11-01597-f002:**
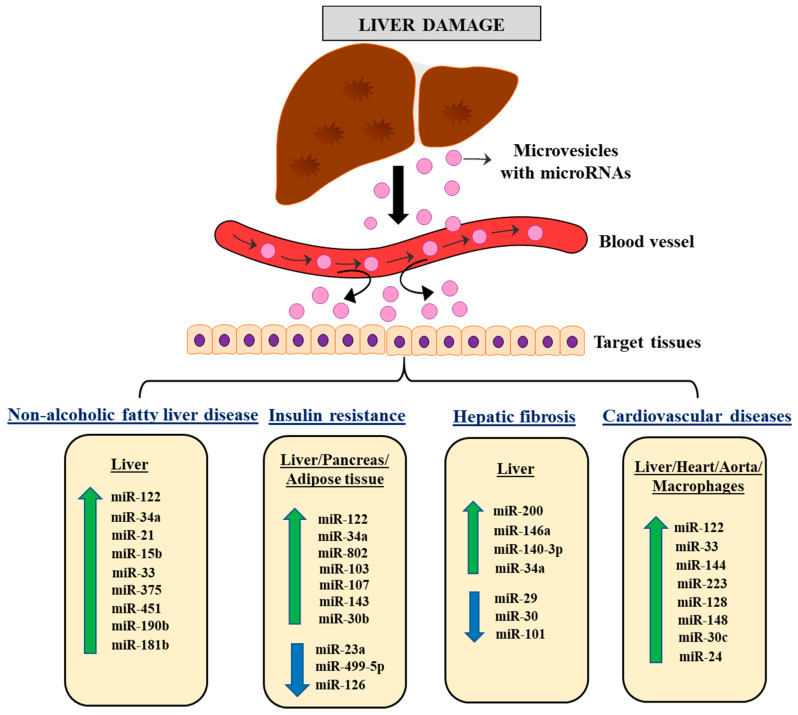
Schematic representation showing the importance of liver dysfunction in the modulation of the expression pattern of different miRNAs, potentiating inflammatory mechanisms and the associated pathophysiological complications responsible for organ damage and consequent disease progression via organ cross-talk.

## Data Availability

Not applicable.

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
