# Peer review of "Insight into the Inter-Organ Crosstalk and Prognostic Role of Liver-Derived MicroRNAs in Metabolic Disease Progression"

_biomedicines, 2023, doi:10.3390/biomedicines11061597_

Round 1

Reviewer 1 Report

This review summarized several liver-derived miRNAs and described how these miRNAs involve in the processes of NAFLD and hepatic fibrosis. In addition, this article also reviewed some liver-derived miRNAs that may mediate the progression of insulin resistance and cardiovascular diseases.

Overall, this article provides useful knowledge for readers who are interested in investigating the related field. The article is well written and organized. There are some suggestions for the authors which may help to provide more in-depth knowledge for this article.

1.    In figure 1, the authors could indicate which miRNAs are involve in the processes such as glucose metabolism, lipid metabolism, oxidative stress, inflammation, fibrosis, etc. in figure 1.

2.    Alternatively, in the 4 tables, the authors could add more information of these liver-derived miRNAs regulated genes how they involve in the pathological processes of NAFLD, hepatic fibrosis, insulin resistance and cardiovascular diseases.

English language is fine. Few errors need to be carefully checked and corrected.  

Author Response

This review summarized several liver-derived miRNAs and described how these miRNAs involve in the processes of NAFLD and hepatic fibrosis. In addition, this article also reviewed some liver-derived miRNAs that may mediate the progression of insulin resistance and cardiovascular diseases.

Overall, this article provides useful knowledge for readers who are interested in investigating the related field. The article is well written and organized. There are some suggestions for the authors which may help to provide more in-depth knowledge for this article.

Thanks for your valuable comments and suggestions.

  1. In figure 1, the authors could indicate which miRNAs are involve in the processes such as glucose metabolism, lipid metabolism, oxidative stress, inflammation, fibrosis, etc. in figure 1.
  2. Alternatively, in the 4 tables, the authors could add more information of these liver-derived miRNAs regulated genes how they involve in the pathological processes of NAFLD, hepatic fibrosis, insulin resistance and cardiovascular diseases.

As suggested by the reviewer, we have included the information about the physiological effects of each miRNA in respective tables. 

Reviewer 2 Report

The manuscript by Goncalves et al., reviews an important topic on liver-derived microRNA in liver and cardiovascular pathophysiology. In general, the review provides good insights on various well-studied microRNAs, how they regulate glucose or lipid metabolism in the liver. Addition of few discussions may improve the overall understanding of microRNA in metabolic diseases.

The title suggests inter-organ cross talk by liver-derived microRNA. But most of the discussion is on how the microRNA affects hepatic metabolism. The review on cardiovascular disease also seems to discuss about the effects of hepatic changes such as cholesterol or lipids and their influence on cardiovascular disease. Several reviews already exist about microRNA’s role in liver physiology.

To justify the title and to provide new insights, the authors should focus on the inter-organ communication between hepatocytes and other organs via microRNA secreted by hepatocytes under various physiopathological states.

A discussion about how microRNA levels is dysregulated in various stages of liver disease would be helpful.

A discussion on the mechanism of microRNA secretion (via exosomes?) and dysregulation in liver disease would be helpful.

A discussion about how microRNA uptake is regulated (in other organs) or is there tissue-specific targeting of microRNA exists in metabolic diseases.

The review talks more about the hepatocellular effects of microRNA on hepatic metabolism and the outcomes of the dysregulated metabolism in cardiovascular function. It is unclear whether circulatory miRNA derived from the hepatocytes are affecting other cell types to induce or progress metabolic diseases.

None

Author Response

The manuscript by Goncalves et al., reviews an important topic on liver-derived microRNA in liver and cardiovascular pathophysiology. In general, the review provides good insights on various well-studied microRNAs, how they regulate glucose or lipid metabolism in the liver. Addition of few discussions may improve the overall understanding of microRNA in metabolic diseases.

 Thanks for your valuable comments and suggestions.

The title suggests inter-organ cross talk by liver-derived microRNA. But most of the discussion is on how the microRNA affects hepatic metabolism. The review on cardiovascular disease also seems to discuss about the effects of hepatic changes such as cholesterol or lipids and their influence on cardiovascular disease. Several reviews already exist about microRNA’s role in liver physiology.

We agree with the reviewer that there are several articles on the role of miRNAs in liver physiology. However, our review focuses on the importance of liver-derived miRNAs in metabolic disease progression viz, NASH, insulin resistance, cardiovascular diseases and hepatic fibrosis, which involves the association liver with other organs including, heart, aorta, macrophages, adipose tissue, and pancreas. The dysregulation of these miRNAs in circulation also signifies the systemic effects that affect different organ systems in the body. Also, through this review, we are trying to highlight the importance of these miRNAs as prognostic markers in the progression of various metabolic diseases. As suggested by the reviewer we have revised the manuscript by including more information about the role of liver-derived miRNAs in other target organs and included a section in the table showing the physiological effects of each miRNA. In the sections, NAFLD and liver fibrosis, we have included the details of previous studies which demonstrate the progressive changes of the miRNAs in various stages of liver pathology. In the sections, insulin resistance and cardiovascular diseases, we have included the role of miRNAs in other target organs. We have modified fig 2 accordingly. Hence our review consolidates the significance of liver-derived miRNAs in organ cross-talk that mediates the progression of various metabolic diseases and highlights their importance as prognostic biomarkers.

To justify the title and to provide new insights, the authors should focus on the inter-organ communication between hepatocytes and other organs via microRNA secreted by hepatocytes under various physiopathological states.

We have revised the sections, insulin resistance and cardiovascular diseases, by including more information about the role of liver-derived miRNAs in other target organs, such as heart, aorta, fat, macrophages, and pancreas. Also, the dysregulation of these miRNAs in circulation also signifies the systemic effects that affect different organ systems in the body. We have modified fig 2 accordingly.

A discussion about how microRNA levels is dysregulated in various stages of liver disease would be helpful.

We have included the details of previous studies which demonstrate the progressive changes of miRNA expression in various stages of liver diseases in sections, NAFLD and liver fibrosis.

A discussion on the mechanism of microRNA secretion (via exosomes?) and dysregulation in liver disease would be helpful.

We have included a section about ‘Biogenesis and release of miRNA from liver’ in which we have discussed exosome-mediated-miRNA secretion.

A discussion about how microRNA uptake is regulated (in other organs) or is there tissue-specific targeting of microRNA exists in metabolic diseases.

We have included a section about ‘Biogenesis and release of miRNA from liver’ in which we have discussed exosome-mediated-miRNA update by target cells. Specific targets for each miRNA are also discussed in respective sections and also presented in tables.

The review talks more about the hepatocellular effects of microRNA on hepatic metabolism and the outcomes of the dysregulated metabolism in cardiovascular function. It is unclear whether circulatory miRNA derived from the hepatocytes are affecting other cell types to induce or progress metabolic diseases.

We have revised the sections, insulin resistance and cardiovascular diseases, by including more information about the role of liver-derived miRNAs in other target organs, such as heart, aorta, fat, macrophages and pancreas.

Reviewer 3 Report

This review describes the role of miRNAs in metabolic diaseases, including NASH

The topic has been described a number of times regarding miRNA and NASH previously; the relatively new issue is the combination of NASH, insulin resistance, cardiovascular diseases and hepatic fibrosis

The study is well-written

The tables are all fine and easy for an overview

My comments:

I really miss a Discussion section or a summary section – even though Figure 2 is good

Where do the authors suggest that the next clinical steps should be taken?

Author Response

This review describes the role of miRNAs in metabolic diseases, including NASH

The topic has been described a number of times regarding miRNA and NASH previously; the relatively new issue is the combination of NASH, insulin resistance, cardiovascular diseases and hepatic fibrosis

The study is well-written

The tables are all fine and easy for an overview

Thanks for your valuable comments and suggestions.

My comments:

I really miss a Discussion section or a summary section – even though Figure 2 is good

As suggested by the reviewer, we have revised the conclusion section.

Where do the authors suggest that the next clinical steps should be taken?

As suggested by the reviewer, we have included information about future clinical steps in the conclusion section.

Reviewer 4 Report

The manuscript is well written, with a correct sequence in the information illustration and the level is appropriate to readership. The topic is very current.  Recent studies show that circulating liver derived miRNAs are playing an important role in diagnosis and prognosis in NAFLD and NASH. The prognostic role of microRNAs in metabolic diseases progression is very intrigue.

Minor revision

In the introduction section, authors should better describe, even if in a synthetic way, the biogenesis of microRNAs

The authors should also mention that since it is very difficult to localize the tissue of origin of miRNAs, except for hepatocyte-enriched miR-122, it turns out to be a captivating challenge to explain the disease association to microRNAs regulation.

 Minor editing of English language required

Author Response

The manuscript is well written, with a correct sequence in the information illustration and the level is appropriate to readership. The topic is very current.  Recent studies show that circulating liver derived miRNAs are playing an important role in diagnosis and prognosis in NAFLD and NASH. The prognostic role of microRNAs in metabolic diseases progression is very intrigue.

Thanks for your valuable comments and suggestions.

Minor revision

In the introduction section, authors should better describe, even if in a synthetic way, the biogenesis of microRNAs

We have included a section about ‘Biogenesis and release of miRNA from liver’

The authors should also mention that since it is very difficult to localize the tissue of origin of miRNAs, except for hepatocyte-enriched miR-122, it turns out to be a captivating challenge to explain the disease association to microRNAs regulation.

As suggested by the reviewer, we have revised the manuscript by mentioning the difficulty in localizing the origin of miRNAs. However, we are mainly focusing on the association of miRNAs secreted from the liver with other organs and their significant role in metabolic disease progression.

 Minor editing of English language required

Revisions has been made in the manuscript to improve the English language.

Round 2

Reviewer 2 Report

The authors have addressed my concerns.